# Response of NDVI and SIF to Meteorological Drought in the Yellow River Basin from 2001 to 2020

Jie Li [1,2,3,4], Mengfei Xi [1,2,3,4], Ziwu Pan [1,2,3,4], Zhenzhen Liu [1,2,3,4], Zhilin He [1,2,3,4] and Fen Qin [1,2,3,4,*]

1 Key Laboratory of Geospatial Technology for the Middle and Lower Yellow River Regions, Ministry of Education, Henan University, Kaifeng 475004, China
2 College of Geography and Environmental Science, Henan University, Kaifeng 475004, China
3 Henan Industrial Technology Academy of Spatio-Temporal Big Data, Henan University, Kaifeng 475004, China
4 Henan Technology Innovation Center of Spatial-Temporal Big Data, Henan University, Kaifeng 475004, China
* Correspondence: qinfen@henu.edu.cn; Tel.: +86-135-0378-5302

**Abstract:** Understanding the response of vegetation to drought is of great significance to the biodiversity protection of terrestrial ecosystem. Based on the MOD13A2 NDVI, GOSIF, and SPEI data of the Yellow River Basin from 2001 to 2020, this paper used the methods of Theil–Sen median trend analysis, Mann–Kendall significance test, and Pearson correlation analysis to analyze whether the vegetation change trends monitored by MODIS and GOSIF are consistent and their sensitivity to meteorological drought. The results showed that NDVI and SIF increased significantly ($p < 0.001$) at the rate of $0.496 \times 10^{-2}$ and $0.345 \times 10^{-2}$, respectively. The significant improvement area of SIF (66.49%, $p < 0.05$) is higher than NDVI (50.7%, $p < 0.05$), and the spatial distribution trend of vegetation growth monitored by NDVI and SIF is consistent. The negative value of SPEI-12 accounts for 65.83%, with obvious periodic changes. The significant positive correlation areas of SIF-SPEI in spring, summer, and autumn (R > 0, $p < 0.05$) were 7.00%, 28.49%, and 2.28% respectively, which were higher than the significant positive correlation areas of NDVI-SPEI (spring: 1.79%; summer: 20.72%; autumn: 1.13%). SIF responded more strongly to SPEI in summer, and farmland SIF was significantly correlated with SPEI (0.3424, $p < 0.01$). The results indicate that SIF is more responsive to drought than NDVI. Analyzing the response of vegetation to meteorological drought can provide constructive reference for ecological protection.

**Keywords:** NDVI; SIF; meteorological drought; correlation analysis; Yellow River Basin





## 1. Introduction

Drought is a natural disaster with high frequency, wide range of influence, and serious losses in the world [1–4]. Under the background of global warming, the drought trend will be strengthened due to the lack of precipitation in a long time and expansive space [5–7]. The increasing drought event has seriously threatened the biodiversity of terrestrial ecosystem [8,9]. Because drought occurs frequently in most climatic zones, drought research has been a concern for scholars [10,11].

In the context of global climate change, drought not only significantly reduces the productivity of vegetation, but also causes large-scale death of vegetation. As an important part of terrestrial biosphere, vegetation plays an irreplaceable role in regulating the global carbon balance, maintaining the stability of global climate and water resources, but it is vulnerable to drought [12,13]. Drought is considered to be one of the important factors affecting vegetation growth and net primary productivity of terrestrial ecosystems [13,14]. Different drought indices were used for drought monitoring, including the Standardized Precipitation Evapotranspiration Index (SPEI) [15], the Palmer Drought Severity Index (PDSI) [16], the Standardized Precipitation Index (SPI) [17]. However, SPEI is obtained by quantifying precipitation, potential evaporation, and temperature. Compared with PDSI

and SPI, SPEI consideration of drought-related climatic factors has the characteristics of multi-timescale [18–20]. Therefore, studying the response characteristics of vegetation to drought is the key basis for analyzing the impact of drought and predicting the drought risk, which is of great significance to the biodiversity protection of terrestrial ecosystem [21].

The developed remote sensing technology is used to monitor the changes of vegetation cover. EVI (Enhanced Vegetation Index), NDVI (Normalized Difference Vegetation Index) and SIF (Sun/Solar-induced Chlorophyll Fluorescence) have been gradually applied to evaluate the response of vegetation growth to drought [22,23]. SIF is a new remote sensing method developed in recent years to study vegetation photosynthesis, and it is one of the main tasks of the European Space Agency's future Earth exploration program. Compared with traditional optical reflection vegetation remote sensing, SIF has great advantages in vegetation stress detection and carbon cycle research [24,25]. Jiao et al. [26] evaluated the correlation between SIF and four commonly used drought indexes of SPI, SPEI, TCI, and PDSI in different climate regions in the U.S. mainland. They believe that SIF is highly sensitive to meteorological drought in arid areas, SIF of different vegetation types has different response to meteorological drought, and farmland is more sensitive to short-term drought. Tian et al. [23] used NDVI and SIF from 2009 to 2018 to study the impact of extreme drought events on the growth of Australian vegetation. The study found that the average NDVI and SIF values in 2018 were significantly lower than the average values in the past decade. Abnormal drought greatly affected the growth of farmland and grassland, and farmland SIF was more sensitive to drought. Song et al. [25] used SIF and Traditional Vegetation Index (NDVI and EVI) to study the high temperature stress faced by winter wheat in Northwest India. Its research shows that the sudden rise of temperature since March 2010 has significantly affected wheat growth, and the yield predicted by SIF satellite observation has significantly decreased, and the response is earlier than that of NDVI and EVI. In the context of climate change, an in-depth understanding of the impact of drought on vegetation is particularly needed, which is conducive to restricting land carbon absorption and scientific climate policy-making in the future.

The Yellow River Basin (YRB) is located in the arid, semi-arid, and semi-humid areas in northern China, with a total drainage area of $79.5 \times 10^4$ km$^2$. It is the fifth largest river in the world and the second largest river in China [27]. YRB has a vast territory and significant differences in climate exist in the different regions, and it is an area significantly affected by climate change [28]. YRB is an important ecological barrier and important economic zone in China, and plays a decisive strategic role in the ecological protection and high-quality development of the Yellow River Basin. In addition, YRB has diverse landforms, climates, and vegetation types. Vegetation plays a vital role in maintaining the stability of regional and global ecosystems [29,30]. YRB drought has seriously increased in frequency and detriment. Evaluating the impact of drought on vegetation dynamics is necessary, which will provide reference suggestions for ecological protection and biodiversity construction.

In recent years, the response of YRB vegetation dynamics to drought has great significance. However, most studies only use the traditional vegetation index to analyze vegetation growth [31–33], and SIF data that are more sensitive to climate are rarely used. It is unclear whether the trend of vegetation change monitored by NDVI and SIF in the Yellow River Basin is consistent, and the sensitivity of NDVI and SIF to drought is uncertain. Therefore, analyzing the response of NDVI and SIF to drought in the Yellow River Basin has become absolutely imperative for drought assessment and management.

## 2. Materials and Methods

### 2.1. Study Area

The Yellow River Basin (YRB) is located in northern China (Figure 1), between 95°53′–119°05′ E and 32°10′–41°50′ N, and is the second largest river in China. It originates in the Bayan Kara mountains of the Qinghai Tibet Plateau and flows through nine provinces and regions including Qinghai, Sichuan, Gansu, Ningxia, Inner Mongolia, Shaanxi, Shanxi, Henan, and Shandong, with a total drainage area of $79.5 \times 10^4$ km$^2$ [31]. YRB stretches

across four geomorphic units from west to east: Qinghai Tibet Plateau, Inner Mongolia Plateau, Loess Plateau, and Huang Huai Hai Plain. YRB belongs to continental climate, and the climate of different regions in the basin is significantly different. Farmland, grassland, and woodland are the main land-use types in YRB [32].

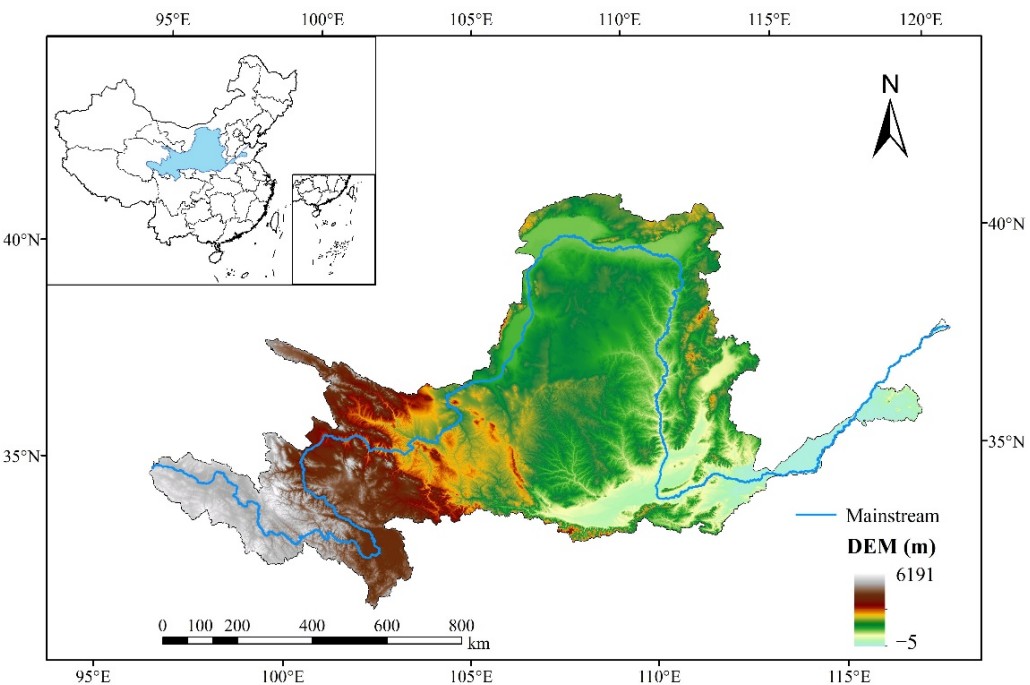

**Figure 1.** Location of the Yellow River Basin, China, and elevation.

## 2.2. Datasets

### 2.2.1. MODIS Dataset

This study used the MOD13A2 NDVI dataset from the Terra satellite from 2001 to 2020 (https://earthdata.nasa.gov/, accessed on 10 June 2021). MOD13A2 is synthesized every 16 days, and the spatial resolution is 1 km. The remote sensing data are synthesized into monthly values by using the maximum value composite method (MVC). NDVI values range from −1 to 1, and negative values indicate that the ground is covered by clouds, water, snow, etc. Due to the influence of aerosols, field of view, clouds, solar altitude angle, and other conditions, MOD13A2 NDVI remote sensing image contains mutation points and noise, consequently Savitzky Golay (S-G) filtering method is used to reduce or eliminate this influence [34]. In order to be consistent with the resolution of GOSIF, the NDVI is resampled to 0.05° using the nearest neighbor method.

### 2.2.2. GOSIF Data

Sun/solar-induced chlorophyll fluorescence (SIF) can be used to describe photosynthesis of vegetation. The spatial-temporal resolution of GOSIF dataset used in this study is monthly and 0.05°, and the time range is 2001–2020 (https://globalecology.unh.edu/data/GOSIF.html, accessed on 20 June 2021). The dataset is developed through data-driven use of discrete OCO-2 SIF soundings, remote sensing data from the Moderate Resolution Imaging Spectroradiometer (MODIS), and meteorological reanalysis data [35]. GOSIF is highly correlated with the GPP of 91 FLUXNET sites ($R^2 = 0.73$, $p < 0.001$), and its spatial-temporal distribution is consistent with the SIF data aggregated directly from OCO-2 SIF. Compared with the coarse-resolution SIF that was directly aggregated from OCO-2 soundings, GOSIF has finer spatial resolution, globally continuous coverage, and a much longer record [35]. GOSIF is useful for assessing terrestrial photosynthesis and ecosystem function and benchmarking terrestrial biosphere and earth system models [35,36].

### 2.2.3. SPEI (Standardized Precipitation Evapotranspiration Index)

SPEI (standardized precision evaporation index) is an index representing the degree of drought obtained by quantifying precipitation, potential evapotranspiration, and temperature. It has the characteristics of multiple time scales. SPEI has been extensively used in the study of vegetation response to drought, providing reliable information for drought research [15,37]. SPEIbase v2.7 of the global SPEI database provides long-term reliable information for drought monitoring worldwide. The spatial-temporal resolution is monthly and 0.5°, and the time range is from January 1901 to December 2020 (https://spei.csic.es/spei_database_2_7, accessed on 10 August 2021). SPEIbase v2.7 calculates potential evapotranspiration based on FAO-56 Penman-Monteith model, which is superior to the estimated results of Thornthwaite PET [21]. Therefore, we used the SPEI data from 2001 to 2020 to identify the spatio-temporal variation trend of drought. Here, the drought represented by SPEI is divided into five categories as shown in Table 1:

**Table 1.** Five level classification of drought based on SPEI.

| Drought Class | SPEI Value |
|---|---|
| Normal | $-0.5 < \text{SPEI}$ |
| Middle drought | $-1.0 < \text{SPEI} \leq -0.5$ |
| Moderate drought | $-1.5 < \text{SPEI} \leq -1.0$ |
| Severe drought | $-2.0 < \text{SPEI} \leq -1.5$ |
| Extreme drought | $\text{SPEI} \leq -2.0$ |

### 2.2.4. Land Cover Type

This study used the land cover data of the European Space Agency with a spatial resolution of 300 m (https://maps.elie.ucl.ac.be/CCI/viewer/download.php, accessed on 15 August 2021). According to the land cover data of the European Space Agency, the different vegetation cover is reclassified into eight types (Figure 2): cropland, grassland, shrubland, woodland, urban areas, bare areas, water, and permanent snow and ice. Farmland, grassland, and woodland are the main land-use types in YRB. The nearest neighbor method was used to resample the land cover type data to the same resolution as GOSIF.

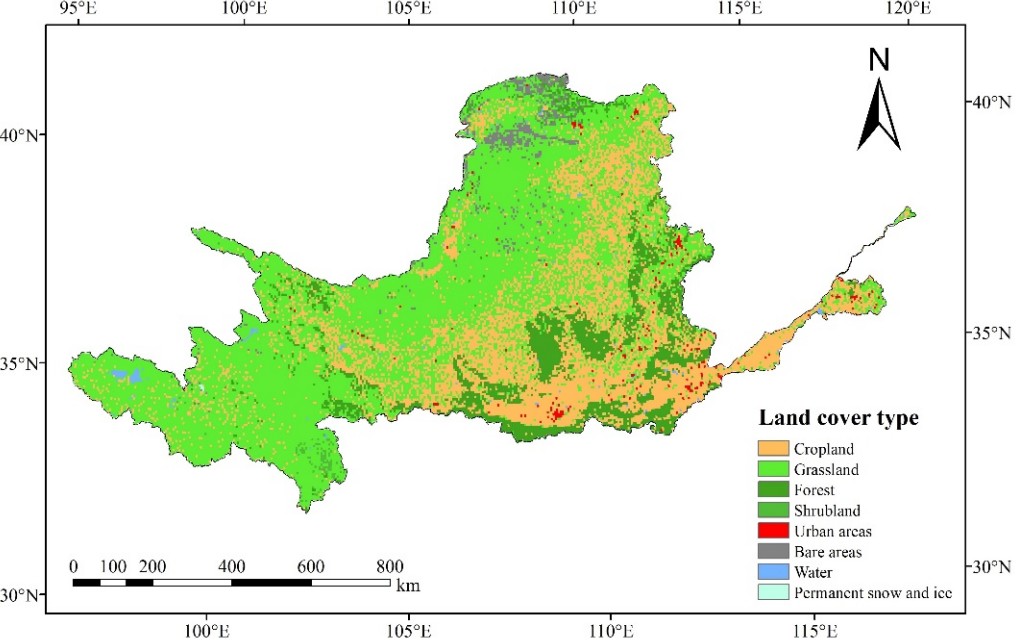

**Figure 2.** Different vegetation types in the Yellow River Basin.

*2.3. Methods*

2.3.1. Theil–Sen and Mann–Kendall (M–K)

The advantage of Theil–Sen median trend analysis is that it does not require samples to obey a certain distribution, and is not disturbed by outliers. It has a strong ability to avoid measurement errors or outlier data [38]. Mann–Kendall is a nonparametric statistical test method used to judge the significance of trends [39]. The calculation formula is as follows:

$$\beta = \text{median}\left(\frac{V_k - V_j}{k - j}\right), \forall\, k < j \tag{1}$$

$$Z_c = \begin{cases} \dfrac{S-1}{\sqrt{var(S)}} & S > 0 \\ 0 & S = 0 \\ \dfrac{S+1}{\sqrt{var}} & S < 0 \end{cases} \tag{2}$$

$$S = \sum_{k=1}^{n-1} \sum_{j=k+1}^{n} sign(V_k - V_j) \tag{3}$$

$$var(S) = \frac{n(n-1)(2n+5) - \sum_{i=1}^{n} t_i(t_i - 1)(2t_i + 5)}{18} \tag{4}$$

$$sign(V_k - V_i) = \begin{cases} 1 & V_k - V_i > 0 \\ 0 & V_k - V_i = 0 \\ -1 & V_k - V_i < 0 \end{cases} \tag{5}$$

where, $\beta$ is the change trend of time series; $k$ and $j$ are the corresponding times; $V_k$ and $V_j$ represent the NDVI or SIF values of the $k$th and $j$th years respectively. If $\beta > 0$, it indicates an increasing trend, and vice versa. At a given significance level $\alpha$, when $|Z_c| > Z_{1-\alpha/2}$ it means that the sequence is in $\alpha$; the level changes significantly. When $|Z_c| > 1.96$, it means that 95% of the significance test has been passed.

2.3.2. Z-Score Method

Z-score is a method often used for data standardization. It standardizes the data based on the mean and standard deviation of the original data to ensure the reliability of the results. SIF has been standardized, and the calculation formula is as follows:

$$A_{b,a} = \frac{SIF_{b,a} - \overline{SIF_b}}{\sigma} \tag{6}$$

where $A_{b,a}$ indicates SIF abnormality in month $b_{\text{th}}$ of year $a_{\text{th}}$. $\overline{SIF_b}$ represents the average SIF of $b_{\text{th}}$ month from 2001 to 2020. $\sigma$ is the standard deviation.

2.3.3. Partial Correlation Analysis

Pearson correlation analysis (R) is widely used to measure the degree of correlation between two variables, with a range between $-1$ and $1$. This study used Pearson correlation analysis to measure the correlation between vegetation and drought pixel by pixel. The calculation formula is as follows:

$$R_{(x,y)} = \frac{\sum_{i=1}^{n}(x_i - \overline{x})(y_i - \overline{y})}{\sqrt{\sum_{i=1}^{n}(x_i - \overline{x})^2}\sqrt{\sum_{i=1}^{n}(y_i - \overline{y})^2}} \tag{7}$$

where $n$ is the length of the time series, $x_i$ is the NDVI or SIF standardized anomaly at the pixel, $y_i$ is the multi-scale SPEI at the pixel, $\overline{x}$ and $\overline{y}$ are the average values of the corresponding research object from 2001 to 2020. $R_{(x,y)}$ is used to quantify the sensitivity of vegetation to drought. The greater the $R_{(x,y)}$, the higher the sensitivity, and vice versa.

## 3. Results

### 3.1. Drought Characteristics of the YRB

#### 3.1.1. SPEI Multi-Time Scale Change Trend

The temporal variation characteristics of meteorological drought were calculated at different time scales (SPEI-1, SPEI-3, SPEI-6, SPEI-9, and SPEI-12) from 2001 to 2020 (Figure 3). The results show that the number of drought events under the five time scales is not identical. SPEI-1 and SPEI-3 have the largest fluctuations, the most frequent fluctuations, and the cyclical law is not obvious. SPEI-9 and SPEI-12 dry and wet events began to show periodic changes, and there was no obvious positive and negative alternation in the short term. The results show that the larger the SPEI time scale, the more negative values. The 1-month scale is 55.83%, the 3-month scale is 60.83%, the 6-month scale is 61.67%, the 9-month scale is 63.33%, and the 12-month scale is 65.83%. Among them, the negative value of SPEI at the 12-month scale is the most (65.83%), and the cyclical change is the most obvious. Therefore, with the gradual increase of SPEI time scale, the fluctuation frequency of SPEI decreases gradually and the periodic change is obvious.

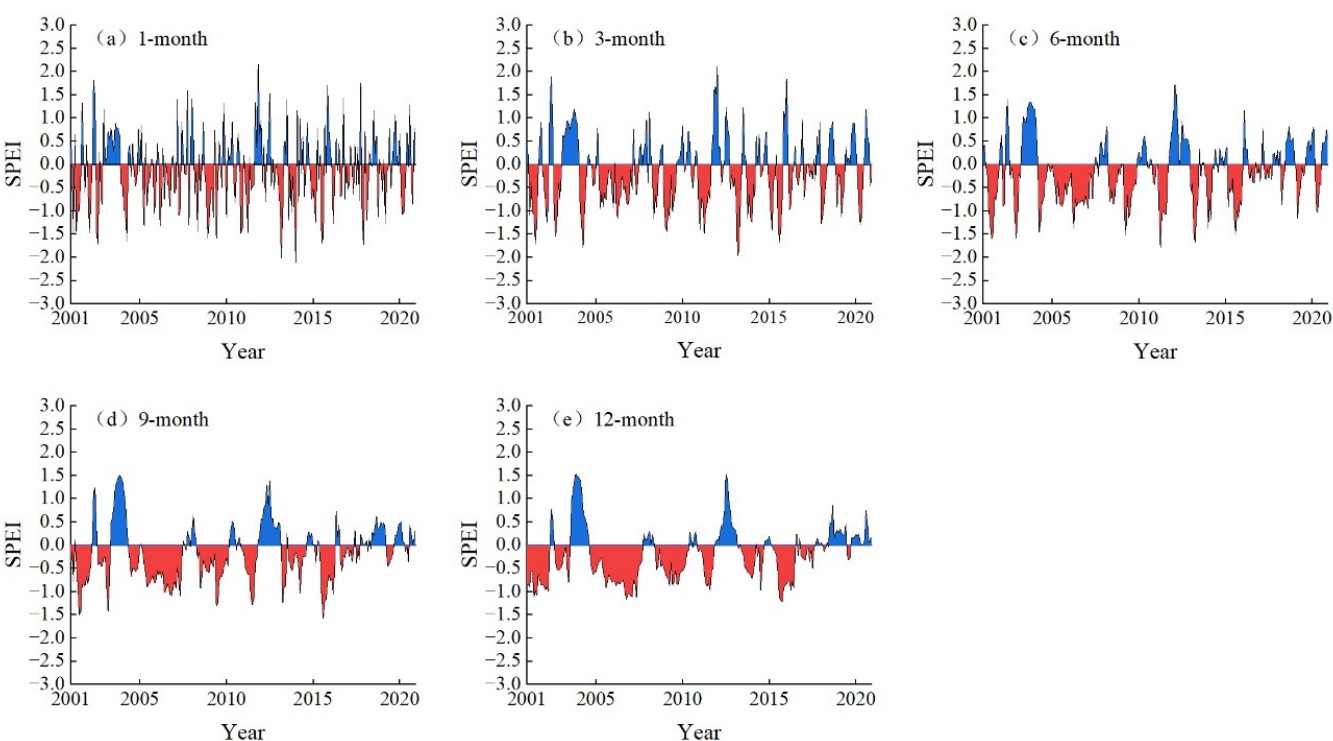

**Figure 3.** Temporal variation of meteorological drought at different time scales in YRB from 2001 to 2020. (**a**) SPEI-1; (**b**) SPEI-3; (**c**) SPEI-6; (**d**) SPEI-9; (**e**) SPEI-12.

Using SPEI-3 and SPEI-12 to analyze the drought temporal variation characteristics of YRB in annual and different seasons, the results are shown in Figure 4. From 2001–2020, the annual SPEI has shown a gradually increasing trend at a rate of 0.0228/a. The fluctuations in the early stage of the year are violent, and the fluctuations in the later stage tend to be stable. SPEI showed a linear downward trend ($-0.2147$/a, $R^2 = 0.0171$) in spring. The fluctuation range was large before 2010 and had a small amplitude after 2010, indicating that YRB showed a gradually drought trend in spring (Figure 4b). SPEI showed an increasing trend in summer and autumn (Figure 4c,d), with an upward rate of 0.0442/a and 0.0061/a respectively. However, the overall fluctuation range in summer is large, and the regularity is not obvious. SPEI in summer reached its minimum in 2015. The rising rate in summer is significantly higher than that in annual, spring, and autumn, indicating that summer presents a wetting trend, which may be related to more rainfall in summer.

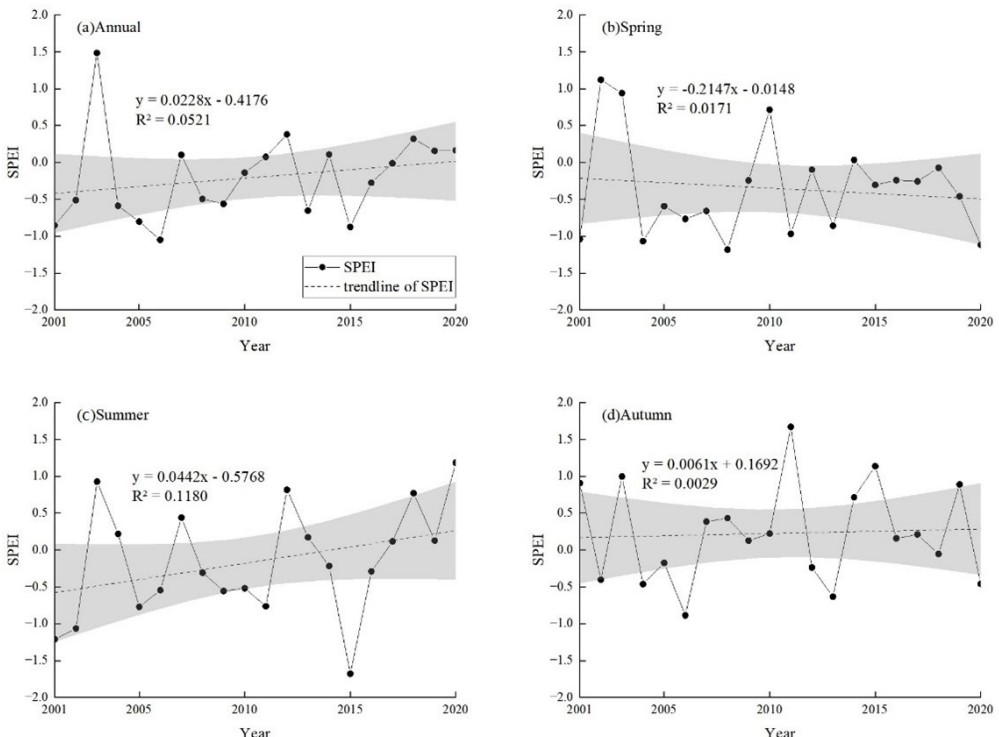

**Figure 4.** Interannual variation of SPEI in YRB 2001 to 2020 (**a**–**d**). (**a**) Annual, (**b**) spring, (**c**) summer, and (**d**) autumn.

### 3.1.2. Spatial Variation Characteristics of SPEI

Pixel by pixel analysis of spatial characteristics of SPEI trend changes and significance of YRB in annual and different seasons from 2001 to 2020 are shown in Figure 5. The results showed that the drought and humidity trends represented by the study area passed the significance test ($p < 0.05$) with fewer pixels. In the annual SPEI trend, the northern part of the study area shows a humid trend, but there is almost no obvious humid area (Figure 5a,e), and the southwest shows a significant drought trend (6.71%, $p < 0.05$), and its main vegetation types are grassland and forest. There is a significant ($p < 0.05$) drying trend in the study area in spring, accounting for 95.78% of the total area of the study area (Figure 5f). There is a trend of humidification in most areas in summer (96.06%) (Figure 5g). Generally speaking, the spatial distribution characteristics of dryness and wetness in spring and summer show opposite trends. In autumn, 52.68% of the area showed a drying trend (Figure 5h), mainly distributed in the north and south of the Yellow River Basin. In spring, SPEI shows an obvious ($p < 0.05$) drying trend, and 5.01% of the area shows a drying trend in summer, which is mainly distributed in the southeast of the study area, indicating that the dry and wet conditions in spring and summer have a great impact on the climate in YRB.

### 3.2. Spatial and Temporal Evolution Characteristics of Vegetation in YRB
#### 3.2.1. Time Variation Characteristics of NDVI and SIF

Based on the climate and vegetation growth characteristics of YRB, this paper defined March to May as spring, June to August as summer, and September to November as autumn. The temporal variation trend of NDVI and SIF in YRB from 2001 to 2020 is shown in Figure 6. NDVI increased significantly ($p < 0.001$) in spring and autumn at the rate of $0.396 \times 10^{-2}$ and $0.373 \times 10^{-2}$, respectively. The average NDVI value of annual and summer is significantly higher than that of spring and autumn. NDVI in annual and summer showed a significant increasing trend at the rate of $0.496 \times 10^{-2}$ and $0.469 \times 10^{-2}$ respectively ($p < 0.001$), and the fluctuation trend was similar. SIF in spring, summer, and autumn showed a significant increasing ($p < 0.001$) trend at the rates of $0.074 \times 10^{-2}$,

$0.122 \times 10^{-2}$, and $0.069 \times 10^{-2}$ respectively. From 2010 to 2012, annual NDVI and annual SIF have a trend of decreasing to obvious increasing. From 2001 to 2020, annual NDVI and annual SIF increased at the rates of $0.5 \times 10^{-2}$ and $0.34 \times 10^{-2}$, respectively.

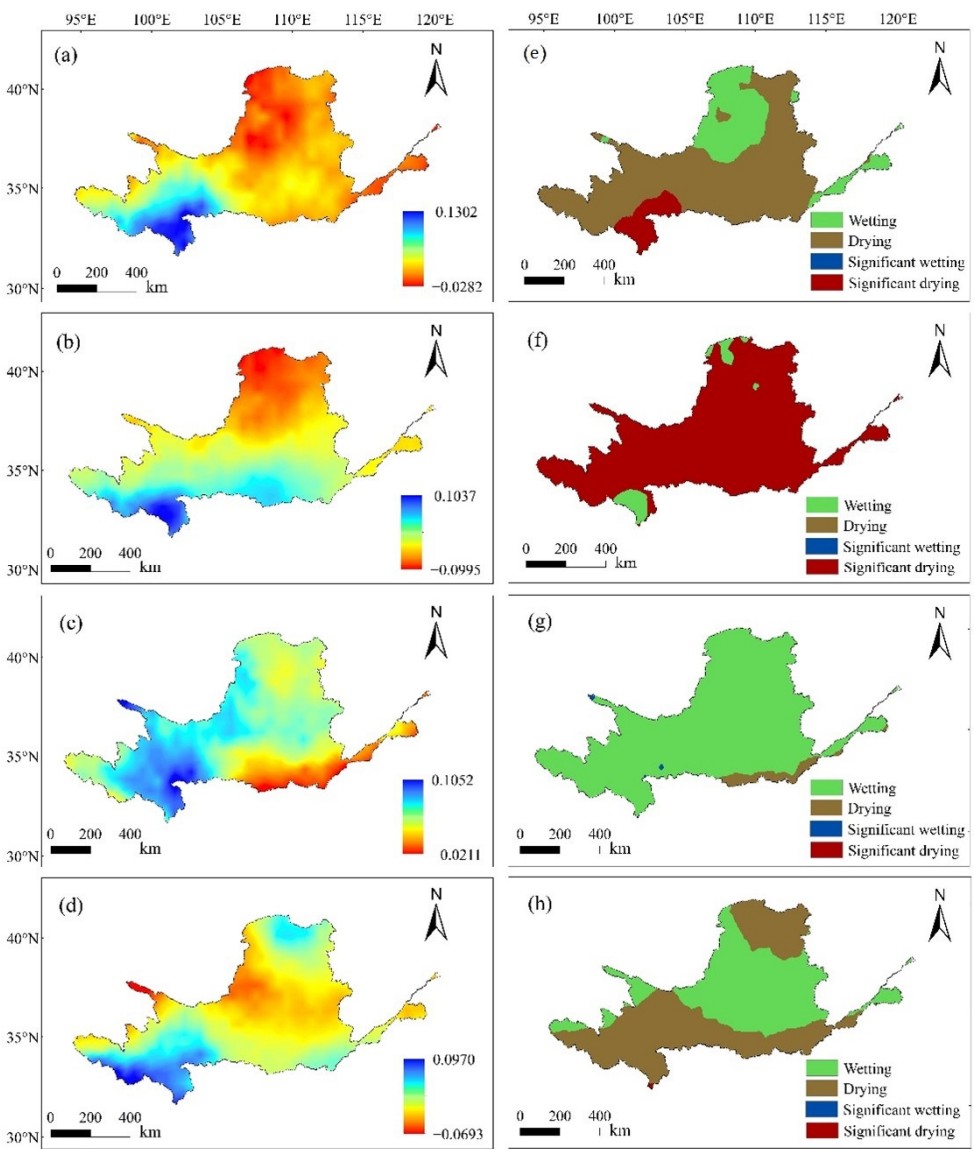

**Figure 5.** Spatial characteristics of SPEI variation trend (**a**: Annual; **b**: Spring; **c**: Summer; **d**: Autumn) and significance (**e**: Annual; **f**: Spring; **g**: Summer; **h**: Autumn) in annual and different seasons ($p < 0.05$).

### 3.2.2. Spatial Evolution Characteristics of NDVI and SIF

In the past 20 years, NDVI and SIF in different seasons in YRB have shown obvious spatial heterogeneity (Figure 7). On the whole, the vegetation increases gradually from north to south throughout the year and in different seasons. The spatial distribution of annual NDVI and annual SIF shows that the south is higher than the north. The pixels with NDVI > 0.5 account for 54.05%, and the pixels with SIF > 0.3 account for 23.35%, which are mainly distributed in the southeast and southwest of the study area. The temperature rises in spring, and the vegetation is in the embryonic stage. NDVI and SIF have monitored that the growth condition in the southeast of the study area is better than that in the north. The vegetation growth in summer is significantly better than that in spring and autumn, with NDVI > 0.5 accounting for 47.74% and SIF > 0.3 accounting for 22.83%, which are mainly distributed in the southwest of the study area. The vegetation growth in autumn monitored

by NDVI and SIF was significantly lower than that in summer. From the spatial distribution of different seasons, the spatial distribution trend of vegetation growth monitored by NDVI and SIF is consistent.

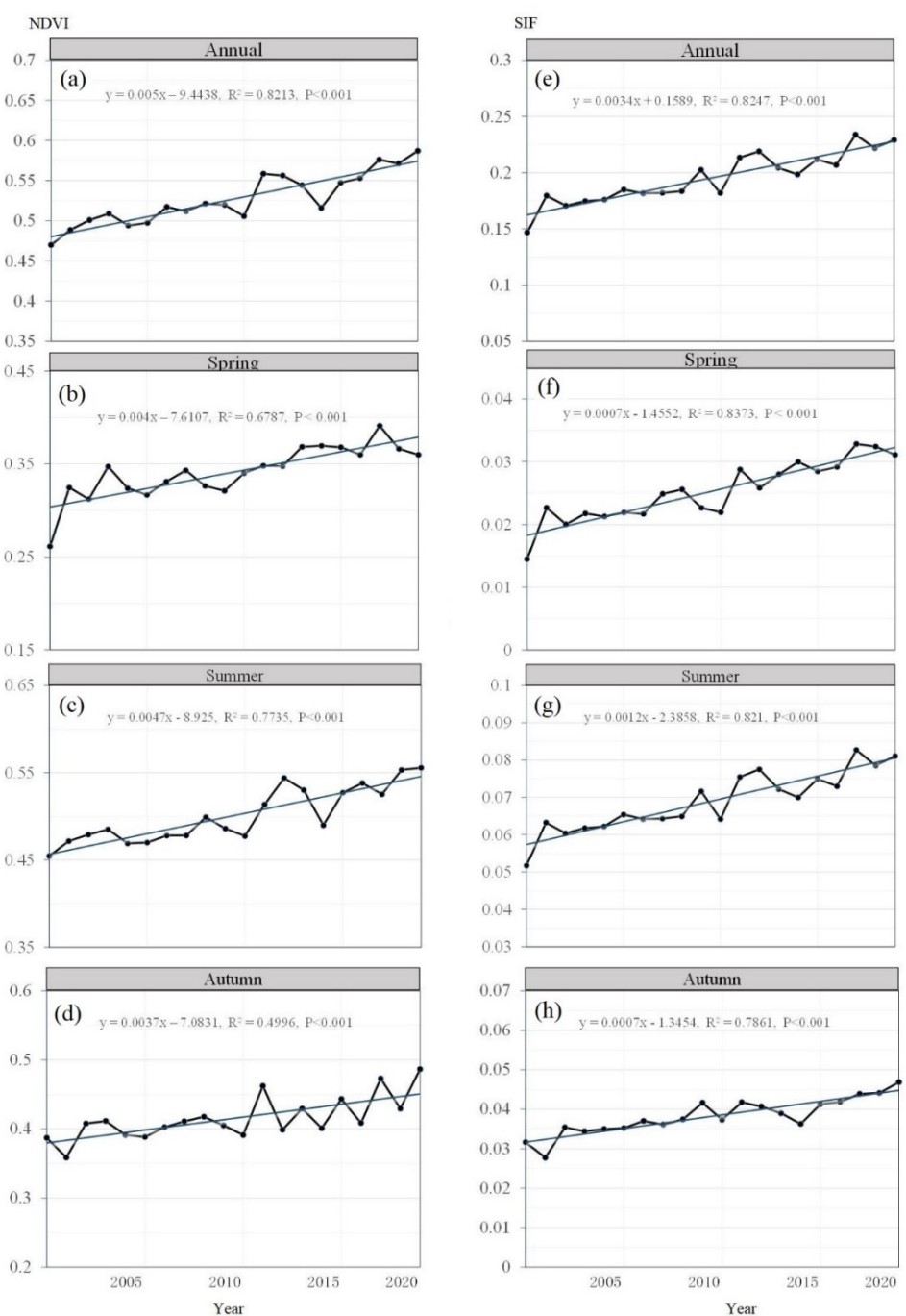

**Figure 6.** The temporal variation trend of NDVI and SIF in YRB from 2001 to 2020 (**a–h**). (**a**) Annual NDVI; (**b**) Spring NDVI; (**c**) Summer NDVI; (**d**) Autumn NDVI; (**e**) Annual SIF; (**f**) Spring SIF; (**g**) Summer SIF; (**h**) Autumn SIF.

Combined with Theil–Sen median trend analysis and Mann–Kendall significance test, the significant change trend of vegetation on the pixel scale in YRB from 2001 to 2020 is obtained (Figure 8). In the past 20 years, the vegetation has shown a significant increasing trend, and the significantly increased area is significantly higher than the significantly degraded area. According to the annual change trend, the significant improvement areas of NDVI and SIF are mainly distributed in the middle of the study area, and the significant

improvement areas of SIF (66.49%, *p* < 0.05) are higher than NDVI (50.7%, *p* < 0.05). In spring, the vegetation monitored by NDVI and SIF has a significant degradation trend in the northern part of the study area, accounting for 2.47% and 1.99% respectively. NDVI monitored that the vegetation showed a significant improvement trend in spring (35.12%, *p* < 0.05), summer (40.20%, *p* < 0.05), and autumn (36.56%, *p* < 0.05). The vegetation monitored by SIF also showed a significant improvement trend in spring (61.75%, *p* < 0.05), summer (66.06%, *p* < 0.05), and autumn (64.39%, *p* < 0.05). The significant change trends of NDVI and SIF are consistent as a whole; however, the significant improvement areas of vegetation monitored by SIF are higher than NDVI.

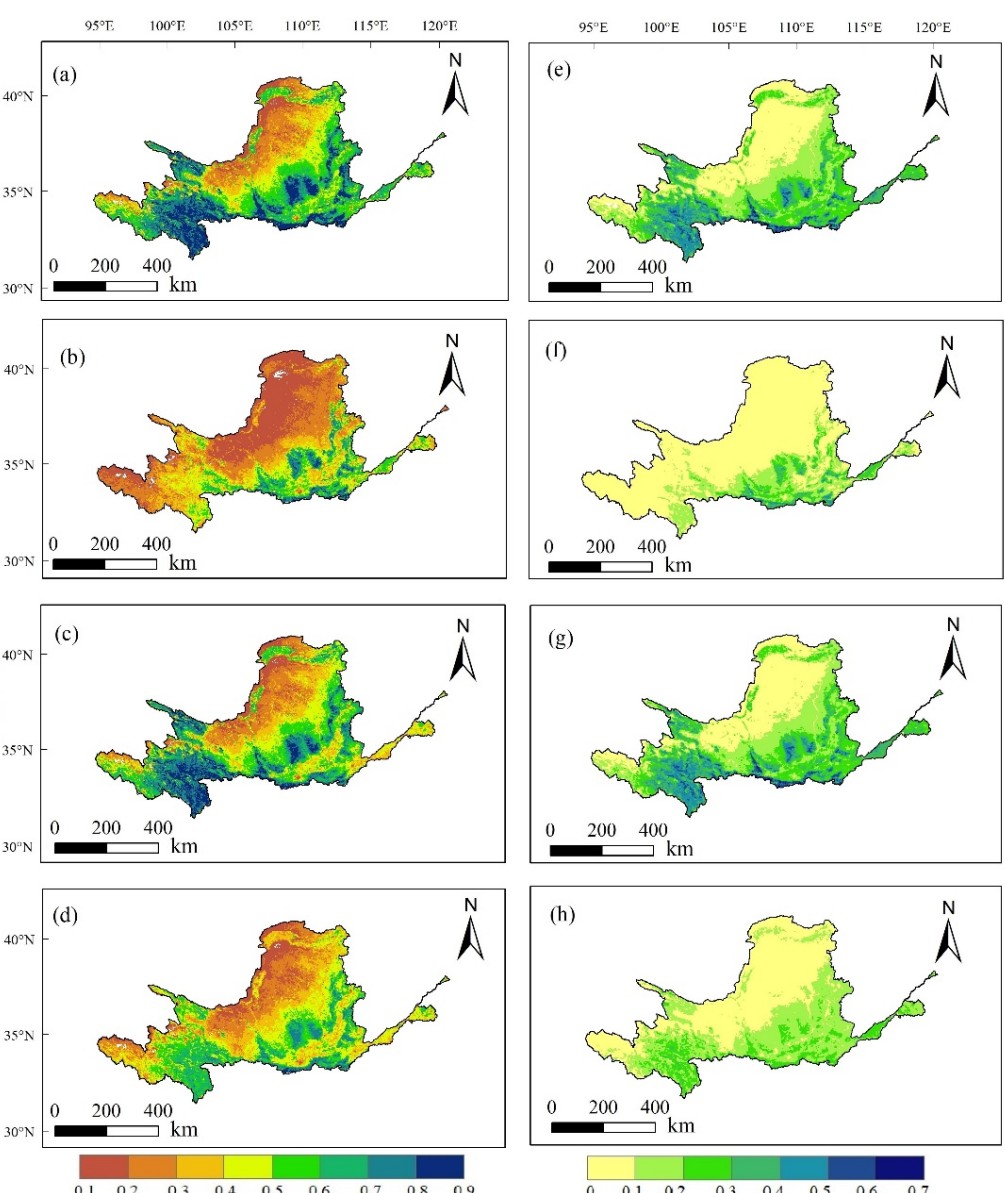

**Figure 7.** Spatial variation of average NDVI and SIF in different seasons (**a–h**). (**a**) Annual average NDVI; (**b**) Spring average NDVI; (**c**) Summer average NDVI; (**d**) Autumn average NDVI; (**e**) Annual average SIF; (**f**) Spring average SIF; (**g**) Summer average SIF; (**h**) Autumn average SIF.

### 3.3. Drought Impacts on Vegetation Growth

3.3.1. Response of NDVI and SIF to Drought

The difference between NDVI and SIF responses to drought from annual and seasonal time scales are shown in Figure 9. Annual NDVI and SPEI are significantly positively correlated (R > 0, *p* < 0.05), accounting for 12.67%, mainly distributed in the northwest. The

area with significant negative correlation (R < 0, *p* < 0.05) between annual SIF and SPEI accounted for 3.25%, mainly distributed in the southwest; significant positive correlation area accounts for 5.92%, which is mainly distributed in the northwest, which is relatively consistent with the spatial distribution of NDVI and SPEI. In spring, SIF and SPEI are significantly positively correlated (7.00%) and mainly distributed in the middle of the study area. In spring, NDVI and SPEI are significantly positively correlated (1.79%) and relatively scattered and irregular. Obviously, the area of SIF and SPEI (28.49%) in summer (R > 0, *p* < 0.05) is significantly higher than that in other seasons, and is higher than that of NDVI and SPEI in summer (20.72%). In autumn, SIF and SPEI showed a significant positive correlation (R > 0, *p* < 0.05), and the area (1.87%) was higher than that of NDVI and SPEI in autumn (1.13%). In autumn, NDVI was significantly negatively correlated with SPEI (R < 0, *p* < 0.05), and the area (19.26%) was significantly higher than that of SIF and SPEI in autumn (1.87%). The regions with significant positive correlation between SIF and SPEI in spring, summer, and autumn (R > 0, *p* < 0.05) were 7.00%, 28.49%, and 2.28% respectively, which were higher than the regions with significant positive correlation between NDVI and SPEI (spring: 1.79%, summer: 20.72%, autumn: 1.13%). Therefore, SIF is more sensitive to drought than NDVI.

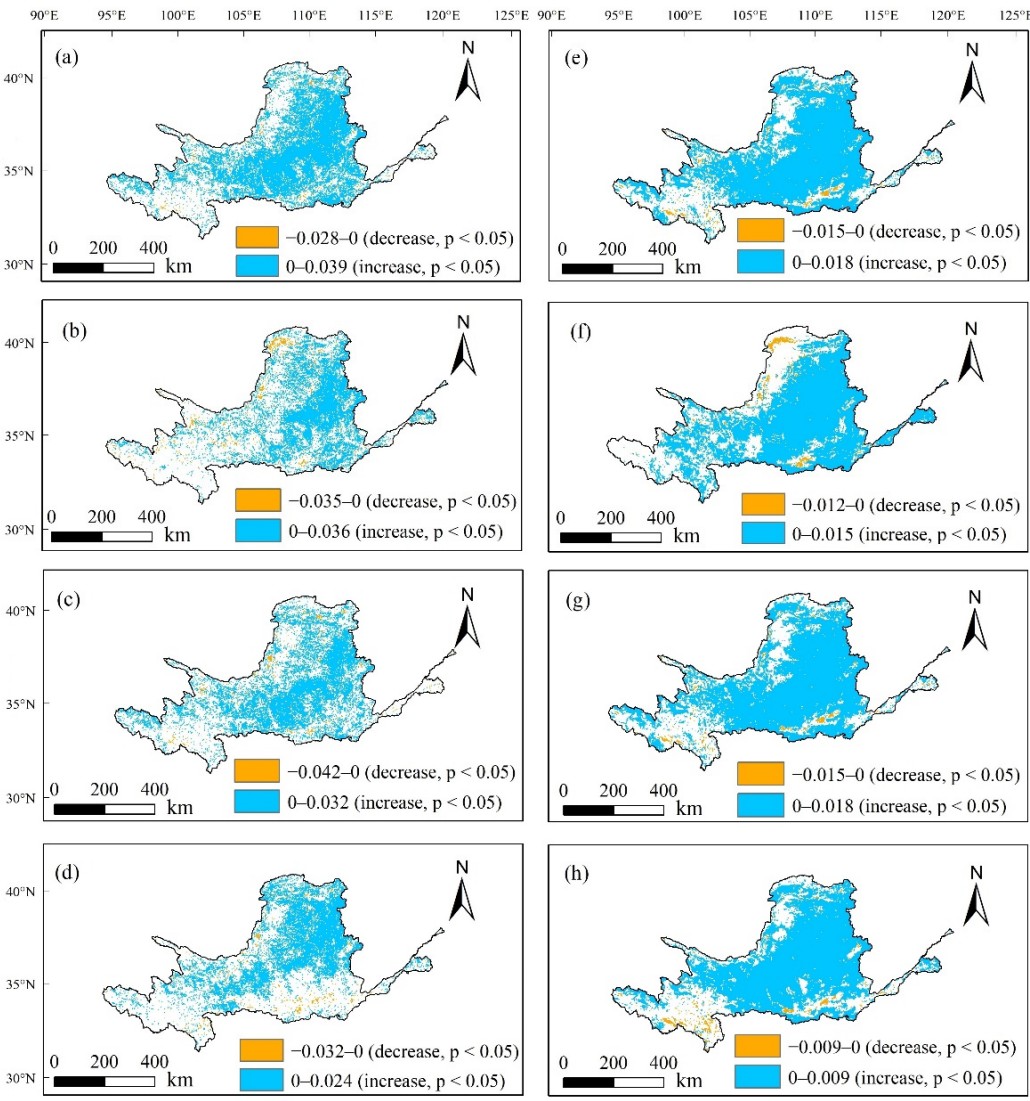

**Figure 8.** Significance test of seasonal vegetation (**a**–**h**). (**a**) Annual NDVI; (**b**) Spring NDVI; (**c**) Summer NDVI; (**d**) Autumn NDVI; (**e**) Annual SIF; (**f**) Spring SIF; (**g**) Summer SIF; (**h**) Autumn SIF.

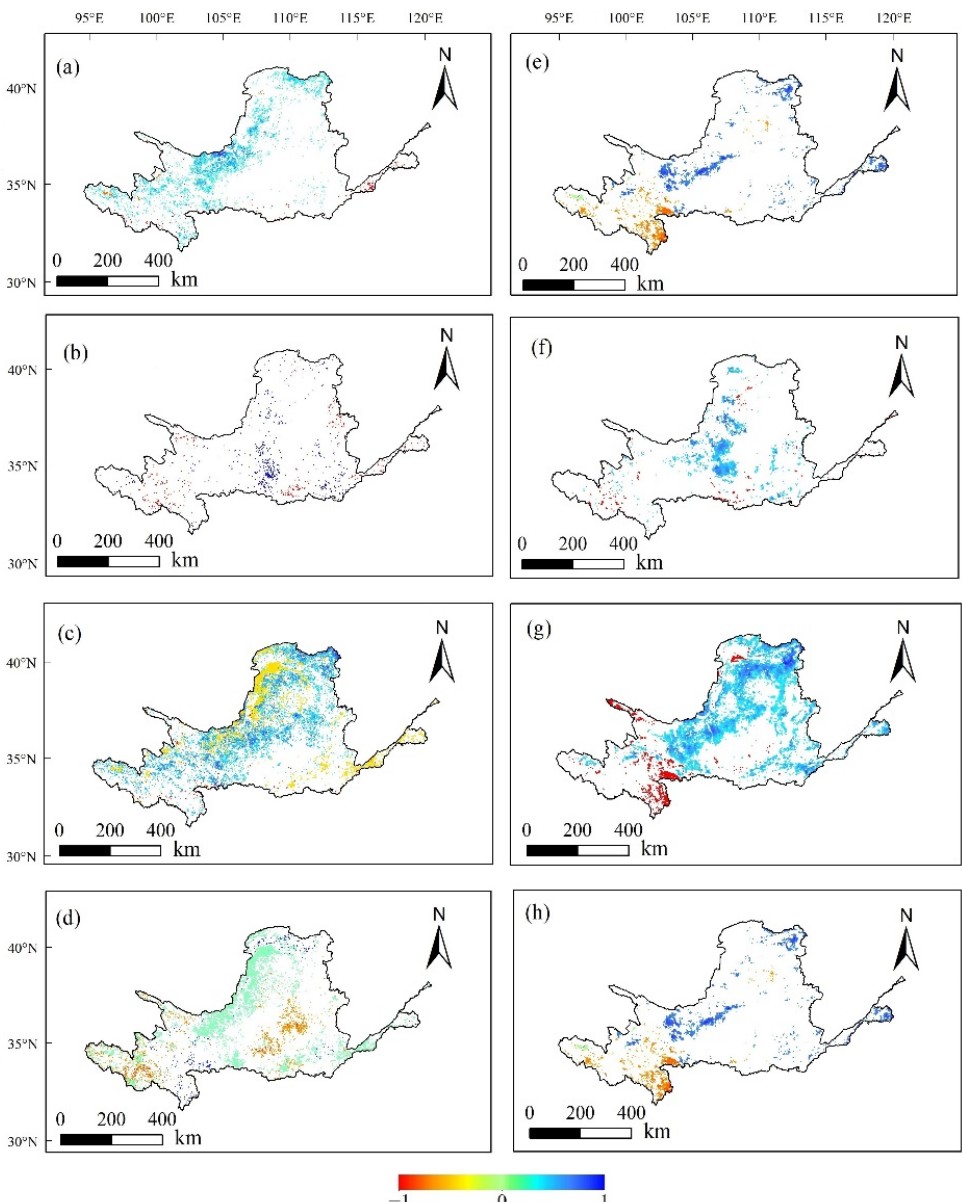

**Figure 9.** Response of NDVI to SPEI in different seasons ($p < 0.05$). (**a**) Annual; (**b**) Spring; (**c**) Summer; (**d**) Autumn. Response of SIF to SPEI in different seasons ($p < 0.05$). (**e**) Annual; (**f**) Spring; (**g**) Summer; (**h**) Autumn.

### 3.3.2. Diverse Droughts Responses by NDVI and SIF

In order to analyze the sensitivity of NDVI and SIF in detecting the impact of drought, typical agricultural areas in YRB were selected to compare the drought response of NDVI and SIF to SPEI (Figure 10). The correlation curves of NDVI, SIF, and SPEI of different land cover types (farmland, grassland and forest) were compared using the scatter plot method. The results show that the linear correlation between SIF and SPEI is greater than that between NDVI and SPEI. The correlation coefficients of SIF and SPEI of farmland and grassland are 0.3424 ($p < 0.01$) and 0.2434 ($p < 0.05$), respectively, as shown in Figure 10d,e. The correlation coefficients of NDVI and SPEI of farmland and grassland are 0.2670 ($p < 0.01$) and 0.1625 ($p < 0.05$), respectively, as shown in Figure 10a,b. The results showed that there was a significant correlation between SIF and SPEI, and the correlation between SIF and SPEI was higher than NDVI, which further showed that SIF was better than NDVI in response to drought. In conclusion, SIF is superior to NDVI in detecting the impact of drought.

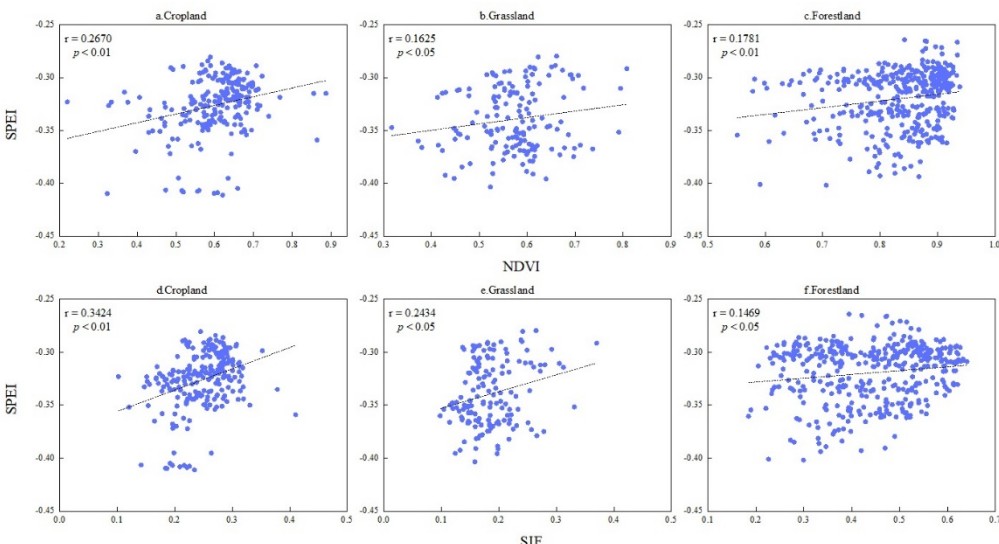

**Figure 10.** Correlation between SPEI and NDVI (**a**) cropland, (**b**) grassland and (**c**) forestland; correlation between SPEI and SIF (**d**) cropland, (**e**) grassland and (**f**) forestland.

## 4. Discussion

### 4.1. Characteristics of Vegetation and Drought

The MODIS dataset is reconstructed with Savitzky Golay (S-G) filter, which can effectively remove outliers and better preserve the continuity of vegetation changes [34]. From 2001 to 2020, the NDVI and SIF of YRB have increased significantly ($p < 0.001$) at the rates of $0.496 \times 10^{-2}$ and $0.345 \times 10^{-2}$ respectively. From the perspective of temporal and spatial variation trends in different seasons, the trend of vegetation growth monitored by NDVI and SIF is consistent, showing a gradually increasing trend. The significant improvement area of SIF (66.49%, $p < 0.05$) was higher than that of NDVI (50.7%, $p < 0.05$). On the one hand, this study confirmed that under the background of global warming, photosynthesis and respiration of vegetation increased, promoting plant growth, and vegetation showed a greening trend [38]. The significant vegetation improvement trend is consistent with the results of other scholars [39,40]. On the other hand, it also shows that SIF is consistent with the vegetation growth monitored by NDVI. SIF indicates vegetation photosynthesis and has the potential to better monitor vegetation growth and response [24].

From 2001 to 2020, the SPEI in spring showed a linear downward trend (0.2147/a, $R^2 = 0.0171$), and the SPEI in spring showed a significant drying trend (95.78%, $p < 0.05$), showing a gradually increasing trend of drought. Because the Yellow River Basin has scarce precipitation in spring, long sunshine duration, rapid temperature rise, rapid decline in surface soil moisture content, and vigorous evaporation, which exacerbated the development of drought [41,42]. The recent sixth assessment report of the United Nations Intergovernmental Panel on Climate Change (IPCC) pointed out that the global warming will reach at least 1.5 °C and the sustained global warming will cause damage to nature, and will aggravate the intensity and frequency of drought, extreme high temperature, sudden changes in ocean circulation, and complex extreme weather events [43]. With the increasing demand of social and economic development for water resources, drought, a natural disaster, has become one of the obstacles to China's agricultural development [3]. The Yellow River is the "Mother River" of the Chinese nation. The ecological environment is fragile and the drought is intensifying, which has seriously affected the local people's life, production, and economic development [44,45].

### 4.2. Sensitivity Analysis of Vegetation Response to Drought

The ecological environment of YRB is fragile and drought events occur frequently. Drought can easily lead to decreased transpiration of vegetation, decreased photosynthetic rate, gradually slowed down or even stagnant growth and development of vegetation,

decreased biomass and even led to plant death [9,46]. NDVI is an important indicator to reflect the growth status and vegetation coverage of vegetation. SIF reflects the photosynthesis of vegetation. NDVI and SIF monitored by remote sensing data can reflect the growth status of vegetation [47]. In order to analyze the difference between NDVI and SIF in detecting the impact of drought in YRB, the study compared the correlation analysis results between NDVI, SIF and SPEI in annual and different seasons (Figure 9). The significant positive correlation areas of SIF-SPEI in spring, summer and autumn (R > 0, $p < 0.05$) were 7.00%, 28.49% and 2.28% respectively, which were higher than the significant positive correlation areas of NDVI-SPEI (spring: 1.79%; summer: 20.72%; autumn: 1.13%). The significant positive correlation area of SIF-SPEI in summer (R > 0, $p < 0.05$) accounted for 28.49%, which was significantly higher than that in other seasons and higher than that in NDVI-SPEI. This result is consistent with the conclusion of sun et al. [48], that is, for temperate grassland, the correlation between SIF and SPEI, PDSI and soil moisture in July and August is much higher than that in other months, and for alpine grassland, the correlation is higher in June and July. Meanwhile, due to high temperature and heat in summer, drought has become one of the environmental stresses faced by plant growth. The high temperature in summer accelerates the physiological activities of plants, thus accelerating transpiration, causing the absorption and supply of roots to be insufficient for the evaporation of plants, resulting in water loss, destroying the water balance of plants, and promoting plants to wither [49,50].

In different land cover types, farmland is more vulnerable to drought than grassland and forest, and farmland SIF has a higher response to SPEI (0.3424, $p < 0.01$). Therefore, we conclude that SIF is more responsive to drought than NDVI, which is consistent with the previous research conclusion, that is, SIF is a more effective drought response index than NDVI [24]. Similarly, song et al. [25] showed that SIF can timely capture the damage to wheat when high temperature stress occurs, half a month earlier than the traditional NDVI and EVI. Leng et al. used multi-source spaceborne SIF to analyze the response of dryland vegetation in Australia to severe extreme drought in the past 20 years, indicating that SIF has a good ability to accurately track the changes in dryland vegetation heterogeneity caused by drought [51]. Tian et al. [23] confirms that SIF is superior to NDVI in detecting the impact of drought, and our conclusion was consistent with him.

Our research also shows that SIF represents vegetation photosynthesis, has the potential to better monitor vegetation growth, and the response of SIF to drought is higher than NDVI [41]. Solar induced chlorophyll fluorescence (SIF) remote sensing is a new remote sensing technology developed rapidly in recent years [52]. The close relationship between SIF and photosynthetic process makes it an effective probe to indicate the Photosynthetic Changes of vegetation. GOSIF describes the photosynthesis of vegetation, continuously covering the world with high spatial resolution. Therefore, GOSIF has broader application prospects and greater potential in vegetation monitoring, global terrestrial carbon cycle, drought stress, carbon neutralization, etc. [35].

*4.3. Limitations*

This study analyzed the temporal and spatial variation trend of vegetation and drought in YRB, and analyzed the response of NDVI and SIF to meteorological drought using Pearson correlation analysis method. However, correlation only indicates the degree of correlation between two variables, and does not represent the causal relationship between them. With the growing interest in SIF emerging satellite products, more satellite products covering the global region will be available in the future. Future research can apply other sensors with higher spatial resolution and time frequency, such as GeoCARB [53], TROPOMI [54] and FLEX [55]. The response of vegetation to drought is related not only to the resistance of vegetation to water stress, but also to the resilience of vegetation at the end of drought events. Vegetation plays a critical role in hydrological cycle, terrestrial carbon cycle and energy exchange [56]. Future research should explore the internal driving mechanism between climate change, drought stress and vegetation change. In the context

of climate change, an in-depth understanding of the impact of drought on vegetation is more important, which can provide a constructive reference for the formulation of land carbon cycle and biodiversity protection policies.

## 5. Conclusions

This study analyzes the response of NDVI and SIF to meteorological drought in the Yellow River Basin from 2001 to 2020. The results show that: from the spatial distribution of different seasons, the significant improvement area of SIF (66.49%, $p < 0.05$) is higher than that of NDVI (50.7%, $p < 0.05$). The spatial distribution of vegetation growth monitored by NDVI and SIF was consistent throughout the annual and different seasons. SPEI-12 has the largest proportion of negative values and has a significant periodic variation. The significant positive correlation areas of SIF-SPEI in spring, summer and autumn (R > 0, $p < 0.05$) were 7.00%, 28.49%, and 2.28% respectively, which were higher than the significant positive correlation areas of NDVI-SPEI (spring: 1.79%; summer: 20.72%; autumn: 1.13%). SIF responded more strongly to SPEI in summer, and farmland SIF was significantly correlated with SPEI (0.3424, $p < 0.01$). The results indicates that SIF was superior to NDVI in detecting drought impacts.

**Author Contributions:** Conceptualization, F.Q. and Z.P.; methodology, J.L. and Z.L.; data curation, J.L. and M.X.; writing—original draft, J.L.; writing—review and editing, J.L. and M.X.; visualization—J.L. and Z.H.; supervision, F.Q. and Z.L. All authors have read and agreed to the published version of the manuscript.

**Funding:** This study was supported by High Resolution Satellite Project of the State Administration of Science, Technology and Industry for National Defense of PRC, grant number 80-Y50G19-9001-22/23; The National Science and Technology Platform Construction, grant number 2005DKA32300; The Key Projects of National Regional Innovation Joint Fund, grant number U21A2014; The Ministry of Education, grant number 16JJD770019; and The Open Program of Collaborative Innovation Center of Geo-Information Technology for Smart Central Plains Henan Province, grant number G202006.

**Institutional Review Board Statement:** Not applicable.

**Informed Consent Statement:** Not applicable.

**Conflicts of Interest:** The authors declare no conflict of interest.

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
