# Peer review of "Response of NDVI and SIF to Meteorological Drought in the Yellow River Basin from 2001 to 2020"

_water, doi:10.3390/w14192978_

Round 1

Reviewer 1 Report

Based on MOD13A2 NDVI, GOSIF and SPEI data, this paper studied the response of NDVI and SIF to meteorological drought in the Yellow River Basin in the past 20 years. The data in this paper is detailed and the results are clear, while the research is not innovative enough. Some issues still need to be improved:

1.     The paper focuses on Yellow River Basin and analysis of its land cover type, but the paper lacks the basis for basin division of the research area, which is recommended to add.

2.     There are many drought indices which have its applicability and disadvantages. Drought index is the key of this research, and the author should justify the use of SPEI.

3.     The description of land cover type in the text is inconsistent with the representation in the figure, it is recommended to unify.

4.     According to the research results of this paper (Fig. 3 and Fig. 4), the Yellow River Basin has shown different trends in the seasons at different time scales in the past 20 years, which indicates drought has an obvious scale effect. However, there is a lack of analysis on the effect of drought scale on vegetation in this paper.

5.     According to the results shown in Figure 6, the change trend of SIF in spring and Autumn is quite different from that of NDVI, but the conclusion drawn in the paper is "From the time change trend, the time change trend of annual SIF is obviously consistent with the fluctuation of annual NDVI.” Is it reasonable? It requires further clarification.

6.     On the whole, the linear correlation between SPEI and SIF and NDVI obtained in this study is not obvious, which may be related to drought indicators or other factors. While this paper lacks the mechanism analysis of the research results.

7.     The discussion needs to be further improved. It is suggested to increase the comparative analysis with the existing research results and reduce the description of the research significance.

8.     The conclusion needs to be further summarized and improved. The conclusion should be a summary of the full text, not a repetition of the results part.

Reviewer 2 Report

Manuscript ID: water-1905265
Title: Response of NDVI and SIF to meteorological drought in the Yellow River Basin from 2001 to 2020
OVERVIEW
The manuscript analyses the vegetation change trends monitored by MODIS and GOSIF and test the sensitivity to meteorological drought, using the MOD13A2 NDVI, GOSIF and SPEI data of the Yellow River Basin from 2001 to 2020.
GENERAL COMMENTS
The subject matter is actual, interesting and within the scope of the Journal Water.
The manuscript complies with the journal template and is well structured.
The English spelling and grammar can be improved.
As for the rest, I have a few suggestions. Please read the specific comments.
In conclusion, I believe this manuscript is interesting and worthy of publication after minor changes.
SPECIFIC COMMENTS
Line 22: where reads “The negative value of SPEI at the 12-month scale is the most (65.83%), and the cyclical change is the most obvious.”, please correct the sentence.
Line 28: where reads “Monitor the spatial and temporal changes and sensitivity of vegetation and drought, and provide reference suggestions for ecological protection.”, please correct the sentence.
Line 56: where reads “Jiao et al.”, please include the reference number.
Line 110: where reads “The range of values is -1-1, and”, maybe 1 to -1. Please correct.
Line 220: where reads “YRB will show a trend of gradually moistening in the future”, please explain and justify further.
Line 395: where reads “sun et al.,”, please include the reference number.
Line 408: where reads “song et al. (2018)”, please include the reference number.

Round 2

Reviewer 1 Report

The author has revised and explained according to the review comments. Overall, I think the article is ready for publication